# Molecular Docking and Dynamics Simulation Revealed Ivermectin as Potential Drug against *Schistosoma*-Associated Bladder Cancer Targeting Protein Signaling: Computational Drug Repositioning Approach

**DOI:** 10.3390/medicina57101058

**Published:** 2021-10-03

**Authors:** Arif Jamal Siddiqui, Mohammad Faheem Khan, Walid Sabri Hamadou, Manish Goyal, Sadaf Jahan, Arshad Jamal, Syed Amir Ashraf, Pankaj Sharma, Manojkumar Sachidanandan, Riadh Badraoui, Kundan Kumar Chaubey, Mejdi Snoussi, Mohd Adnan

**Affiliations:** 1Department of Biology, College of Science, University of Hail, Hail 2440, Saudi Arabia; walidsabrimail@gmail.com (W.S.H.); arshadjamalus@yahoo.com (A.J.); badraouir@yahoo.fr (R.B.); snmejdi@yahoo.fr (M.S.); drmohdadnan@gmail.com (M.A.); 2Department of Biotechnology, Era’s Lucknow Medical College, Era University, Lucknow 226003, India; faheemkhan35@gmail.com; 3Molecular Parasitology and Immunology Division, CSIR-Central Drug Research Institute, Lucknow 226031, India; manish55552@gmail.com (M.G.); pspks6154@gmail.com (P.S.); 4Department of Medical Laboratory Sciences, College of Applied Medical Sciences, Majmaah University, Al Majmaah 11952, Saudi Arabia; s.jahan@mu.edu.sa; 5Department of Clinical Nutrition, College of Applied Medial Sciences, University of Hail, Hail 2440, Saudi Arabia; amirashrafy2007@gmail.com; 6Program in Cellular and Molecular Medicine, Boston Children’s Hospital, Boston, MA 02115, USA; 7Department of Oral Radiology, College of Dentistry, University of Hail, Hail 2440, Saudi Arabia; smanojk68@gmail.com; 8Section of Histology-Cytology, Medicine Faculty of Tunis, University of Tunis El Manar, Tunis 1017, Tunisia; 9Department of Biotechnology, Academic Block VI, GLA University, Mathura 281406, India; kundan2006chaubey@gmail.com; 10Laboratory of Genetics, Biodiversity and Valorization of Bio-Resources (LR11ES41), University of Monastir, Higher Institute of Biotechnology of Monastir, Avenue Tahar Haddas BP74, Monastir 5000, Tunisia

**Keywords:** *Schistosoma*, bladder cancer, repurposing drugs, ivermectin, arteether, praziquantel, molecular docking, molecular dynamics simulation

## Abstract

Urogenital schistosomiasis is caused by *Schistosoma haematobium* (*S. haematobium*) infection, which has been linked to the development of bladder cancer. In this study, three repurposing drugs, ivermectin, arteether and praziquantel, were screened to find the potent drug-repurposing candidate against the *Schistosoma*-associated bladder cancer (SABC) in humans by using computational methods. The biology of most glutathione S-transferases (GSTs) proteins and vascular endothelial growth factor (VEGF) is complex and multifaceted, according to recent evidence, and these proteins actively participate in many tumorigenic processes such as cell proliferation, cell survival and drug resistance. The VEGF and GSTs are now widely acknowledged as an important target for antitumor therapy. Thus, in this present study, ivermectin displayed promising inhibition of bladder cancer cells via targeting VEGF and GSTs signaling. Moreover, molecular docking and molecular dynamics (MD) simulation analysis revealed that ivermectin efficiently targeted the binding pockets of VEGF receptor proteins and possessed stable dynamics behavior at binding sites. Therefore, we proposed here that these compounds must be tested experimentally against VEGF and GST signaling in order to control SABC. Our study lies within the idea of discovering repurposing drugs as inhibitors against the different types of human cancers by targeting essential pathways in order to accelerate the drug development cycle.

## 1. Introduction

Despite the in advancements of prognosis therapies, bladder cancer is the eleventh most leading urological cancer that is continuously contributing a 3.0% burden to total cancer with an estimated 573,278 new cases and 212,536 deaths worldwide as obtained from the GLOBOCAN database 2020 [1]. Although the recurrence and mortality rate of bladder cancer have decreased in developed countries, it is still a health concern and an unsolved clinical problem in developing and Eastern European countries where one-third of new cases are being reported continuously every year for the last two decades [2,3]. Generally, bladder cancer is caused by the exposure of urothelial cell linings of the bladder to a number of environmental and occupational hazard risk factors including smoking, toxic chemicals and mutagenic agents (polycyclic aromatic hydrocarbons) that are filtered into the urine by the kidneys [4]. Apart from chemical agents, biological factors such as bacterial infections and other parasites also play a key role in predisposing individuals to bladder cancer [5]. *S. haematobium*, a parasitic flatworm that infects more than 100 million people mostly in the developing world, is the causative agent of urogenital schistosomiasis, and it is associated with a high incidence of squamous cell carcinoma of the bladder [6]. *S. haematobium* is released from infected freshwater snails via chronic granulomatous inflammation in the mucosal and submucosal linings of the urinary bladder. This cascade results in squamous cell carcinoma which is more prevalent and recurrent than compared to conventional transitional cell carcinoma of the urinary bladder [7]. Consequently, numerous studies in the Middle-East, Africa and tropic regions of Asia put forward strong pieces of evidences that *Schistosoma*-associated urinary bladder cancer is more commonly caused by *S. haematobium*, a parasite of freshwater snails, than compared to any other parasitic infection [7]. During infection, eggs are deposited in the bladder causing an intense inflammatory reaction. These reactions result in the formation of oxygen and nitrogen free radicals that increase the production of carcinogenic compounds and, thus, result in bladder cancer. A growing amount of evidence also points to angiogenesis playing a key role in SABC as well as in schistosomiasis. This statement may seem to be a paradox, since schistosomes are intravascular parasites that cause damage by destroying the blood vessels. On the other hand, angiogenesis, or the formation of new endothelial sprouts from pre-existing postcapillary venules, is a well-known characteristic of inflammatory diseases, wound repair and provides support for tumor progression [6,8].

The biological functions of GSTs are to detoxify the xenobiotic compounds and biosynthesis of important biomolecules through catalyzing glutathione by attacking sulfhydryl groups to electrophilic centers found in lipids and fatty acids in order to make them more water-soluble [9]. As depicted in Figure 1, the GSTs act as endogenous enzymes in order to synthesize and transport the metabolites across various cells. In mammalian cells, they also work as antioxidant to scavenge the free radicals at low concentration, whereas the overexpression of GSTs result in angiogenesis and apoptosis cascade in neoplastic cells [10]. Moreover, the biosynthesis of prostaglandin D2 (PGD2), which is catalyzed by GST enzyme, is of vital importance in the life cycle of *S. haematobium* because it penetrates the host’s tissue and, thus, inhibits host immune response [11]. In such a way, *Schistosoma* GST facilitates the entry of parasite into the human cell, where it shows easy fecundity and pathogenicity. *Schistosoma* GST could, thus, be strikingly associated with bladder cancer in humans, and its inhibition could be involved in the anti-fecundity effect of *S. haematobium* as well as in the treatment of SABC [12]. Previous studies have shown that reduced glutathione (GSH) levels and GST activity can reduce the covalent binding of the activated forms of well-known carcinogens, e.g., benzo(a)pyrene, NDMA, aflatoxin B1, DNA and other macromolecules. The higher activity of GST with some polycyclic aromatic hydrocarbons in many human tissues suggests that GST may play an important role in the detoxification of lung carcinogens. Keeping in line with these findings, the inhibition of GST activity in bladder cancer and in *Schistosoma*-infected human bladder cancer tissues may potentiate the binding of alkylating moieties, resulting from bioactivation of N-nitrosamines, with DNA or proteins. The inhibition of GST activity might be due to the presence of certain types of N-nitrosamines in the urine of schistosomal patients, since these compounds are well-known to decrease such activity [13,14]. Since the late 1960s, several pharmaceutical agents such as niridazole, hycanthone, amoscanate and mitrofonate were used for the treatment of schistosomiasis, but now they are prohibited because they exhibit toxic effects. Currently, since it possesses high-spectrum anti-parasitic activity, praziquantel is the drug of choice and is extensively used against all known species of schistosomiasis parasite [15]. Several clinical studies showed the safe and effective use of praziquantel as it is well absorbed in the gut, has easy metabolism in the liver and is excreted out from the body by urine [16].

The molecular pathways underlying the evolution of different cancers have been extensively elucidated, and investigations are now focused on exploring the receptor proteins and their targeted chemical agents for treatment of this disease [17,18]. Amongst them, vascular endothelial growth factor (VEGF), an important angiogenesis activating protein, facilitates the proliferation of endothelial cells to form the new blood vessels during angiogenesis induced by pathological conditions such as solid tumor growth and, in turn, nourishes them by delivering oxygen and nutrients [19]. A large number of proangiogenic factors including VEGF, basic fibroblast growth factor, hepatocyte growth factor and angiogenin are often overexpressed in tumors. Several studies have indicated that angiogenic activators play an important part in the growth and spread of tumors [20,21]. Upon immuno-histochemical examination, the VEGF family and their receptors were found to be expressed in about half of the human cancers investigated [20]. These factors are known to affect the prognosis of adenocarcinomas that have developed in the uterine cervix, endometrium, ovary and stomach. In addition, a significant correlation between the expression of VEGF and prognosis has been described in colorectal, breast, lung cancer, head and neck squamous cell carcinoma, Kaposi sarcoma and malignant mesothelioma [20,21].

Numerous studies have shown that VEGF is found at a significant level in the urine of bladder cancer patients, which is noted to be an independent prognostic indicator correlating to the known grade of cancer, vascular invasion and metastasis [19]. Evidence-based studies show that angiogenesis-dependent VEGF inhibition is the most relevant therapy for combating the cancer disease, which has resulted in the development of several anti-VEGF agents by targeting the VEGF pathway (e.g., aflibercept), VEGF receptors (e.g., ramucirumab) and the inhibition of tyrosine kinase signaling [22]. The inhibition of VEGF factor by anti-VEGF agents results in a reduction in blood vessel formation and reduced interstitial pressure around the cancer cells together with increased delivery of other chemotherapeutic agents at the target sites [23]. Thus, chemicals agents such as ivermectin and arteether could be viable therapeutic agents together with praziquantel against pathological conditions including schistosomiasis and bladder cancer in individuals. These drugs can also be used in general for *Schistosoma* non-associated bladder cancer treatment [24,25,26].

Currently, the main issues of the drugs’ availability against parasitic diseases and cancer are limited, and some drugs are resistant against these diseases. However, it is well known that ivermectin is generally used for the treatment of different parasitic infection such as strongyloidiasis, trichuriasis, ascariasis, filariasis, onchocerciasis and other diseases [27]. Ivermectin possesses promising activity against positive sense single stranded RNA viruses such as dengue, Zika and the more recently identified SARS-CoV-2 via inhibiting viral propagation and replication processes [28]. On the other hand, for the last decades, artemisinin and its derivatives (e.g., arteether) are used to treat malarial fever and a variety of inflammatory conditions such as asthma, pancreatitis, systemic lupus erythematosus and hemorrhagic shock [29]. Arteether can effectively inhibit a number of bacteria, fungi and wide range of DNA and RNA viruses including SARS-CoV-1 and SARS-CoV-2 [30,31]. Moreover, in vitro and in vivo studies demonstrated that artemisinin has anticancer potential by having pro-apoptotic, anti-proliferative, anti-angiogenesis and anti-metastatic effects in breast, glioma, colon, lung, ovarian, pancreatic, renal cell and leukemia cancer cell lines [32]. In addition, several studies revealed the inhibition of proliferation, metastasis and angiogenic processes in diverse cancer cells by the use of ivermectin via targeting multiple signaling pathways including EGFR/ERK, PAK1 kinase, Akt/mTOR, P2X7/P2X7 WNT-TCF SIN3 domain NS3 DDX23 helicase and chloride channels [26,33].

Many parasitic drugs (e.g., praziquantel, albendazole and ivermectin) are available in the market that possess enough efficacies to inhibit or block different pathways, including cancer signaling. In depth studies are required to combat microbial infections and associated diseases by employing different methods such as repurposing and computational analysis [34]. For this purpose, in this study, we evaluated the efficacy of ivermectin and arteether in the bladder cancer caused by *S. haematobium* following a repurposing approach analyzing by molecular docking and MD simulations. Thus, the aim of this study is to examine the efficacy and possible mechanisms of the antimalarial and anti-parasitic drugs in bladder cancer caused by *S. haematobium* using various conventional and advanced techniques including molecular docking and dynamics, wherein the most suitable cascade for targeting will be identified. This study was further designed with a possible assumption supported by various evidence that VEGF and GST pathways are somehow directly or indirectly involved in SABC and can possibly be targeted by these two drugs, which are already approved and under clinical trial for their anticancer efficacy.

## 2. Materials and Methods

### 2.1. Chemical Structures of Ligand

The 3D chemical structures of ligands, namely ivermectin (PubChem CID: 6321424), arteether (PubChem CID: 3000469) and praziquantel (PubChem CID: 4891), were retrieved from the PubChem database (https://pubchem.ncbi.nlm.nih.gov/) (accessed on 15 May 2021) website and saved in SDF format. The geometries of each ligand were cleaned up, and energy minimization was performed by applying MMFF94 (Merck Molecular Force Field) using the LigPrep module of the Schrödinger^®^ software (Schrödinger, München, Germany) (accessed on 15 May 2021) prior to docking. Later on, the structures of ligands were assessed for the correction of bond length and bond angles followed by the addition of missing hydrogen atoms. The conversions of two-dimensional into three-dimensional conformations and their conversion from SDF format to MOL file format were performed with the help of OpenBabel (Open Babel Development Team, Pittsburgh, PA, USA) molecular software. These files were converted into PDB file (.pdb) in order to be visualized by using Biovia^®^ Discovery Studio 3.0 Visualizer (Biovia, San Diego, CA, USA) [35]. Finally, the optimized conformations of ivermectin, arteether and praziquantel were used further for molecular docking and MD simulation studies.

### 2.2. Target Receptor (or Protein) Preparation

The 3D crystal structures of target proteins for VEGF (PDB ID: 1VPF) with 2.50 Å resolution and GST (PDB ID: 1GTA) with 2.40 Å resolution were retrieved from PDB (https://www.rcsb.org/structure/) (accessed on 15 May 2021) in PDB format. Prior to molecular docking, the refinement and energy minimization of protein structures were performed by removing hetero atoms, water molecules and other residual ions followed by the addition of hydrogen atoms and inserting Kollman charges. All the structures were visualized with the help of Biovia^®^ Discovery Studio 3.0 Visualizer (Biovia, San Diego, CA, USA). The energy minimization of the receptors was performed by using a default RMSD value of 0.30 Å and OPLS3e force field methods. Finally, protein structures were refined for molecular docking study by using AutoDock 4.2.6 tool (Scripps Research, San Diego, CA, USA) [36,37].

### 2.3. Pharmacophore Modelling and Design

Prior to molecular docking study, ligands were subjected to pharmacophore modelling establishment after corrections of ligand structure using Ligand Scout 4.4.5 module (InteLigand, Vienna, Austria) [38]. Pharmacophore modelling explored the typical pharmacophore features of a ligand that take part in reactivity against the target protein receptors. Numerous pharmacophore features such as aromatic ring, H-bond donor, H-bond acceptor and hydrophobicity were organized into a map to form a protein–ligand complex.

### 2.4. Molecular Docking

Molecular Docking analyses were performed three times independently by AutoDock Vina Server using the Lamarckian genetic algorithm on ivermectin/arteether/praziquantel-VEGF protein models. Prior to docking studies, the active sites were predicted via online server Prosite (https://prosite.expasy.org/index.html) (accessed on 15 May 2021), and then the grid boxes for both protein receptors were generated. The grid boxes of 126 × 80 × 126 Å (x, y and z) and 85 × 100 × 100 Å (x, y and z) dimensions with binding cavities centered at (x = 16.308, y = 7.235, z = 6.235) Å and (x = 12.078, y = 45.373, z = 46.309) Å, respectively, were automatically generated by using the Auto Grid program. The lowest binding energies and predicted inhibition constants were obtained from docking log files. The set parameters such as Van der Waals and the electrostatic forces of AutoDock (Scripps Research, San Diego, CA, USA) were used for calculating interactions in the molecular docking studies. The protein–ligand complexes and their molecular interactions were visualized by Pymol 2.4.0 (Schrödinger, München, Germany) and Ligplot Plus 2.2.4 software (European Bioinformatics Institute, Cambridgeshire, UK), respectively.

### 2.5. Molecular Dynamic (MD) Simulation

The poses of ivermectin–VEGF and ivermectin–GST complexes possessing the lowest binding energies were selected to evaluate the stability of complexes. MD simulation was performed using the Desmond module of Schrodinger LCC computational software (Schrödinger, München, Germany) for 100 ns time of frame. Prior to MD simulation, the complex structures were optimized under the Schrodinger interface. The conditions for MD simulation were optimized by using isotropic Martyna–Tobias–Klein barostat and Nose–Hoover thermostat methods at a constant temperature of 300K with an atmospheric pressure of 1 bar and a cut-off range at 9 Å. The Coulombic interaction using Particle Mesh Ewald (PME) and the solvation of complex were performed by using the TIP3P water model with a system builder. The selected basic parameters during the simulation included root mean square deviation (RMSD), root mean square fluctuation (RMSF) and ligand properties such as the likely radius of gyration, intermolecular hydrogen bond, molecular surface area, solvent accessible surface area, polar surface area, etc., to evaluate the binding interactions within the ivermectin–VEGF and ivermectin–GST complexes.

## 3. Results

### 3.1. Pharmacophore Modelling

Pharmacophore modelling is a key strategy of ligand-based virtual designing of proposed drug molecules that involves extracting chemical features of 3D structures of a set of ligands binding with protein receptors at specific sites. The interacting chemical features of ligand–protein complexes can assist in generating pharmacophore models with higher similarity than compared to existing drug molecules [39,40]. In the present study, the pharmacophore models were established by using ivermectin, arteether and praziquantel ligands in complex with VEGF and GST proteins. These models revealed that ivermectin possessed thirteen H-bond acceptor atoms, three H-bond donor atoms and nine hydrophobic moieties, whereas arteether was found to have five H-bond acceptor atoms and four hydrophobic moieties. In addition, praziquantel showed the presence of two H-bond atoms, one hydrophobic moiety and one aromatic ring. Thus, these features of ligands discussed here have the capabilities to act as antagonist/agonist against the targeted proteins (Figure 2).

### 3.2. Molecular Docking

As assessed by literature surveys, VEGF and GST pathways are potential target sites for the prevention of various human cancers [41,42,43]. Additionally, it is also known that anti-parasitic drugs such as ivermectin, arteether and praziquantel have also been used against different types of human cancers, but their molecular mechanism is still un-known. In order to obtain deep insights of the mode of action, a molecular docking study was performed to analyze the binding modes of the drugs molecules against different types of human cancer via downregulating VEGF and GST signaling with the help of AutoDock Vina software (Scripps Research, San Diego, CA, USA). The docking results of drug molecules against receptor proteins are depicted in Table 1.

The docking results of drug molecules against receptor proteins are depicted in Table 1. Our docking results revealed good docking scores in terms of binding energies. The results demonstrated that higher negative values of binding energy would be the best binding affinity of ligands within binding pockets of receptor proteins. Ivermectin potentially docked with both protein receptors VEGF and GST by showing the best binding energies. Ivermectin exhibited the highest binding energies −9.2 kcal/mole and −9.0 kcal/mole against VEGF and GST protein receptors, respectively [Table 1]. It binds efficiently with the VEGF receptor through amino acid residues Ile29, Glu30, Thr31, Leu32, Cys57, Gly59, Cys61 and Asp63 of VEGFs possessing both H-bond and hydrophobic bond interactions. The docking of ivermectin with GST receptor displayed three hydrogen bond interactions with amino acid residues Tyr7, Arg108 and Lys125 and eleven hydrophobic interactions with amino acid residues Ile10, Leu13, Leu100, Asp101, Tyr104, Gly105, Ser107, Ile109, Tyr111, Phe122 and Lys125, respectively. Furthermore, the binding energy of arteether was calculated at −5.5 kcal/mole and −6.0 kcal/mole against VEGF and GST receptors, respectively, showing nine molecular interactions with amino acids residues Asp34, Ser50, Gly59, Cys60, Asp63, Glu64, Glu67, Cys68 and Lys107 of VEGF receptor proteins as well as five other molecular interactions with amino acid residues Tyr7, Ser107, Tyr111, Leu13 and Gln204 of GST receptor proteins, respectively. Praziquantel was selected as the reference drug to compare the docking results of ivermectin and arteether, where it docked moderately with VEGF and GST receptors by exhibiting binding energies −6.1 kcal/mole and −6.5 kcal/mole, respectively. Praziquantel showed eight H-bond and hydrophobic bonds interactions with amino acid residues Gly59, Leu32, Ile29, Glu30, Thr31, Arg56, Cys57 and Gly58 of the VEGF receptor. It also displayed six molecular interactions with amino acid residues Trp8, Tyr7, Leu13, Asn54, Tyr111 and Gln207 of the GST receptor. As a result, ivermectin showed better docking scores, whereas arteether displayed weak docking scores than the reference drug praziquantel against both VEGF and GST protein receptors in terms of inhibiting cancer. The three-dimensional (3D) and two-dimensional (2D) mode of binding of drug molecules within the binding pockets of protein receptors are shown in Figure 3, Figure 4 and Figure 5.

### 3.3. Molecular Dynamics (MD) Simulation

In order to analyze the conformational changes and stability of ivermectin within the pockets of protein receptors, an MD simulation study was assessed for 100 ns for ivermectin–VEGF and ivermectin–GST complexes. The RMSD values are generally acclimated to explain the stability of the complex systems. The RMSD values are depicted in Figure 2. The RMSD values of proteins and ligands were calculated as 1.957–8.631 Å and 1.990–15.763 Å for ivermectin–VEGF, whereas 1.026–3.041 Å and 2.596–12.065 Å were calculated for ivermectin–GST complex, respectively. In a trajectory of 100 ns simulation, it was observed that the ligand showed stability with rotational movement from 1 ns to 58 ns in the binding cavity of the ivemectin–VEGF complex. Later on, the ligand changed its position for a while within a 2 ns time frame and achieved stability after 60 ns, which was maintained until 96 ns time of period. Furthermore, the RMSD values for the ivermectin–GST complex were also calculated at 1.026–3.041 Å and 2.596–12.065 Å against protein and ligand, respectively (Table 2). In this complex, the ligand exhibited stability up to 10 ns but started to detach early after 10 ns and then remained in translational movement up to 40 ns. Shortly after, it attained stability in its parental place throughout the entire simulation for 100 ns.

The RMSF value demonstrates the flexibility (positional changes) of the amino acid residues in the presence of the ligand. In the present study, the RMSF values were noted as 0.993–7.538 Å and 0.401–6.916 Å for ivermectin–VEGF and ivermectin–GST complexes, respectively, throughout the simulation period. Surprisingly, ivermectin showed decent flexibility with the amino acid residues of both the proteins VEGF and GST. Among both the complexes, ivermectin–VEGF exhibited higher bond interactions with lesser flexibility of amino acid residues of beta pleated sheet structure. Consequently, the RMSF and RMSD values of both the complexes were found reliable and gave us enough warrant for further analyses.

### 3.4. Protein–Ligand Contact Analysis

As summarized in Figure 6 and Figure 7, simulation analysis was also utilized to calculate the radius of gyration, solvent accessible surface area, molecular surface area and polar surface area in order to obtain additional molecular properties of ligands. The radius of gyration (rGyr) defines the rigidity, flexibility and mobile nature of protein–ligand complexes. The high value of rGyr expresses loose packaging, whereas a low value expresses the compact nature of complex systems. The complex ivermectin–VEGF displayed lower values of rGyr at 5.772–7.321 Å than compared to the complex ivermectin–GST, which showed higher values of rGyr at 5.882–7.373 Å. The solvent accessible surface area (SASA) descriptors describe the protein surface area that is accessible by solvent. The upper value of SASA determines expansion, whereas lower value provides the condense nature of protein. The ivermectin and VEGF protein complex was in an unstable state from 0 to 40 ns, then it achieves a stable state from 40 to 50 ns. However, a higher rise in values was observed at the end of simulation. Comparatively, the ivermectin and GST complexes exhibited stability between 30 and 50 ns, poessesing effective solvent accessible surface area. Molecular surface area (MolSA) and polar surface area (PSA) are useful molecular descriptors that are used to determine transport properties including intestinal absorption and blood–brain barrier (BBB) penetration of a ligand. Ivermectin showed lower values of MolSA and PSA as 699.99–785.11 and 160.604–204.182 in the ivermectin–VEGF complex, displaying higher intestinal absorption and BBB penetration than compared to ivermectin–GST complex where MolSA and PSA values are found at 722.78–783.77 and 164.98–209.46, respectively (Figure 6 and Figure 7).

Hydrogen bonds play a pivotal role in maintaining the conformational integrity of ligands within the active binding sites of protein residues. Thus, simulation trajectories were used for the calculation of hydrogen bonds in both complexes. As shown in the Figure 7, Ref. [32] the ivermectin–VEGF complex forms more hydrogen bonds and water bridges than the ivermectin–GST complex (Figure 6). Consequently, a higher number of hydrogen bonds and water bridges enhance protein–ligand binding capacity. Thus, the ivermectin– VEGF complex is more stable than the ivermectin–GST complex, as was observed in the entire simulation process.

## 4. Discussion

The present study examined the potential use of FDA-approved antimicrobial drugs, namely ivermectin, arteether and praziquantel, to combat human cancers via downregulating VEGF and GST signaling using drug repurposing, molecular docking and MD simulation approaches. In addition to environmental and occupational factors, bladder cancer is mainly caused by *S. haematobium*, a parasite released by infected fresh water. Furthermore, the observations indicated that the expression of GST produced by SH is one of the causes responsible for bladder cancer, with high infectivity rates of schistosomiasis followed by bladder cancer, which makes it a suitable drug target for pharmaceutical studies. Accordingly, many studies have been reported in which the administrations of anti-parasitic drugs, for example, ivermectin and arteether, are observed as a competitive candidate for combating other diseases, especially cancer, because of their involvement in a variety of molecular pathways [44].

Due to efficacy, safety, cost and, indeed, the lack of alternatives, praziquantel has remained the drug of choice for schistosomiasis treatment and transmission control for >40 years [45]. However, praziquantel has drawbacks, including inactivity against juvenile *schistosomes*. Moreover, reliance on a single drug for the treatment of a disease with global public significance of schistosomiasis risks facilitating the development and spread of drug resistance, especially since reduced susceptibility has occurred frequently both in the field and in the laboratory [45,46]. A pressing need for new interventions has arisen, including novel compounds with modes of action discrete from those of praziquantel [46]. Reduced susceptibility to praziquantel has already been widely found in foci of endemicity and is notable in Africa, including Egypt and Senegal. An extremely low cure rate (18%) was reported in Senegal; however, it was suggested that the failure of praziquantel therapy occurs because of factors other than drug resistance, including very intense transmission and the presence of praziquantel-refractory juvenile worms (immature parasites) [46,47].

On the other hand, ivermectin proved to be even more of a ‘wonder drug’ in human health, improving the nutrition, general health and well-being of billions of people worldwide [48]. Ivermectin is a highly effective, broad-spectrum, safe, well tolerated and easily administered drug (a single, annual oral dose). It is used to treat a variety of internal nematode infections, including onchocerciasis, strongyloidiasis, ascariasis, cutaneous larva migrans, filariases, gnathostomiasis and trichuriasis, as well as for oral treatment of ectoparasitic infections such as pediculosis (lice infestation) and scabies (mite infestation) [27,48].

The broad level advantages of ivermectin suggest that it is worth looking into it as a potential new anticancer drug. Ivermectin can selectively inhibit the proliferation of tumors at a dose that is not toxic to normal cells and can reverse multidrug resistances of tumors [26]. It has been widely used in humans for many years, and its various pharmacological properties, including long and short-term toxicological effects and drug metabolism characteristics, are well-known. In addition, ivermectin has also been proven to show good permeability in tumor tissues [26]. Ivermectin has powerful antitumor effects, including the inhibition of proliferation, metastasis and angiogenic activity, in a variety of cancer cells. This may be related to the regulation of multiple signaling pathways and programmed cancer cell death, including apoptosis, autophagy and pyroptosis [26]. Few in vivo and in vitro studies explained the possible mechanism of action of ivermectin [49]. One such example of ovarian cancer study demonstrated that ivermectin was found to cause DNA damage through the induction of double-strand DNA breaks and induces intrinsic apoptosis by disrupting the mitochondrial membrane associated with the upregulation of BAX/BCL-2 and cytochrome C release [49,50]. Similar effects were observed in ivermectin-treated liver cancer. Interestingly, ivermectin induces cell death in leukemia via a different mechanism of action, such as increasing the intracellular level of chloride ions via plasma–membrane hyperpolarization and increasing ROS levels [49,51]. Furthermore, ivermectin also inhibits cell proliferation and induces apoptosis in colon cancer cells by blocking the canonical WNT pathway only on TCF-dependent cell types [52]. In glioblastoma, ivermectin inhibits angiogenesis and deactivates the Akt/mTOR signaling pathway following mitochondrial stress and enhanced ROS levels [49,53]. Therefore, our study was designed with a possible assumption supported by various evidence that VEGF and GST pathways are somehow directly or indirectly involved in SABC and can possibly be targeted by these two wonder drugs already approved and under clinical trial for anticancer efficacy.

Similarly, arteether is another safe and effective drug either alone or in combination with some other antimalarial drugs. No adverse events were found in various studies. As first-line antimalarial medicines, arteether is safe, low-toxic and well tolerable. Furthermore, outcomes from a limited number of clinical trials provide encouraging evidence for their excellent antitumor activities [54,55]. However, some problems such as poor solubility, toxicity and controversial mechanisms of action hamper their use as effective antitumor agents in the clinic. In order to accelerate the use of arteether in the clinic, researchers have recently developed novel therapeutic approaches including developing novel derivatives, manufacturing novel nano-formulations and combining arteether with other drugs for cancer therapy [54,55].

The mechanism of antitumor action of arteether mainly involved in apoptotic cell death has been confirmed by most literature. However, new mechanisms of action in antitumor activity of arteether by affecting non-apoptotic cell death including oncosis, autophagy and ferroptosis were also found [56]. Other multiple hallmark events of cancer development and progression were also affected by arteether, including the suppression of cancer cell proliferation, anti-angiogenesis, anti-cancer metastasis and invasion, induction of cell cycle arrest, disruption of cancer signaling pathway and regulation of tumor microenvironment. Arteether further inhibits the glycolysis capacity in various tumor cells [56,57]. Mi et al. first reported that DHA suppressed glucose uptake and glycolysis in non-small cell lung carcinoma cells and confirmed the effect associated with inhibiting mTOR activity and reducing glucose transporter 1 (GLUT1) expression [58]. Subsequently, Vatsveen et al. observed that ARS decreased glycolysis capacity and mitochondrial respiration capacity in B-cell lymphoma cells, although detailed mechanisms remain to be elucidated [59]. However, the inhibition of glycolysis of DHA was illustrated via inhibiting the PI3K/AKT pathway, downregulating HIF-1α expression and downregulating pyruvate kinase M2. In addition to an effect on glycolytic metabolism, ARS inhibited HCT116 colon cancer cell proliferation by suppressing the fatty acid biosynthetic pathway, mainly downregulating three proteins: acyl-CoA synthetase 5, hydroxyacyl-coenzyme A dehydrogenase and fatty acid synthase [54,59,60].

Despite several studies and treatment options, there are currently no approved therapeutic agent or drug that specifically targets SAB*C*. An exponentially growing number of *in silico* studies by using computational tools have tried to provide molecular data in support of some treatments. The aim of this study was to propose a robust *in silico* protocol that overcame the limitations of classic virtual screening studies. Furthermore, the results of molecular docking and MD simulation for 100 ns time frame confirmed that ivermectin and arteether obstructed bladder cancer by inhibiting VEGF and GST pathways. Ivermectin has good binding energy with both VEGF and GST targets by showing docking scores of −9.2 and −9.0, respectively. In this study, ivermectin-2VPFs in 0–100 ns, Gly59, His86, Gln87, Leu32, Asp34, Phe36, Gln37, Ser50, Cys57, Cys68, Glu73 and His99 residues have a key part in the strong hydrogen interactions that are located in the interaction pocket of VEGF proteins expressing in cancer cells. Moreover, the complex ivermectin–GST is stabilized by H-bonded interactions, which are shown by Tyr7, Trp8, Tyr104, Arg108, Tyr111, Ser112, Gln204 and Gln207 residues in the binding pockets of the GST protein. The values for RMSD (Cα atoms and ligand fit on protein) as well as RMSF (Cα atoms) were consistent with the stability of both the complexes formed during the entire time period of 100 ns MD simulation. In the case of ligand properties, ivermectin shows permissible values of rGyr, MolSA, SASA and PSA during the formation of the ivermectin–VEGF complex. However, this complex has been found to have two water bridges through Asp34 and Glu64 residues of VEGF protein. Furthermore, ivermectin also forms a stable complex in a most favorable conformation with GST protein. In such case, most of the residue Tyr7, Trp8, Tyr104, Arg108, Tyr111, Ser112, Gln204 and Gln207 interacted through hydrogen bonds within the sites located in the conserved motifs of the ivermectin–GST complex. Furthermore, MD simulation indicated that ivermectin resides in the binding pocket of the GST protein within favorable values of RMSD and RMSF for 100 ns time frame. Moreover, the values of ligand properties such as rGyr, MolSA, SASA and PSA were consistent with fit values of drug abilities. Surprisingly, being a stable complex, it has not shown water bridge interactions.

On the other hand, arteether displayed moderate docking scores at −5.5 and −6.0 against VEGF and GST proteins because the lesser number of hydrogen bonds with the residues Asp34, Ser50, Gly59, Cys60, Asp63, Glu64, Glu67, Cys68 and Lys107 for the VEGF receptor and residues Tyr7, Ser107, Tyr111, Leu13 and Gln204 for the GST receptor. The hydrophobic bonds and water bridges were not significant for the arteether while interacting with both the protein receptors. Moreover, a docking study of praziquantel as a reference drug resulted in a less significant docking score than compared to ivermectin, whereas the docking results of praziquantel were more significant when compared to arteether’s results. In addition, pharmacophore modelling suggested that ivermectin has a total of thirteen hydrogen bond acceptors and three hydrogen bond donor sites, followed by five and two hydrogen bond acceptors in arteether and praziquantel ligands, respectively. Moreover, arteether and praziquantel were devoid of hydrogen bond donor sites. Pharmacophore modelling, thus, revealed that ivermectin could be interact strongly with both the receptor proteins via the formation of hydrogen bonds. Therefore, based on our analysis of molecular docking and MD simulation studies, the formed complexes of ivermectin–VEGF and ivermectin–GST have relatively the best results in terms of protein–ligand interaction. In this case, ivermectin would be a suitable drug molecule for targeting VEGF and GST signaling for the treatment of SABC in human. Furthermore, the advantage of focusing on ivermectin and FDA-approved drugs is that the safety issues are all within suitable bounds and are well understood, meaning that it could proceed to clinical trial reasonably quickly against SABC in humans. Comparatively, our results clearly indicated that ivermectin is more potent than arteether against SABC due to the weak binding efficacy of arteether to the targeted receptors.

## 5. Conclusions

In our study, ivermectin and arteether were predicted as potential inhibitors of SABC by taking into account the molecular interactions and their stability during the formation of a protein–ligand complex. Hydrogen acceptor, hydrogen bond donor and water bridges together with conformational changes may be responsible for molecular recognitions and specificities of anti-parasitic drug (e.g., ivermectin, arteether and praziquantel) interaction with VEGF and GST protein signaling in SABC. In order to warrant this study, several validation studies, including computational, in vitro, in vivo and clinical are needed to prove the efficacy of these drugs prior to approval. Moreover, another limitation of this study is that the exact involvement of VGEF and GST pathways in SABC is not fully known. Furthermore, there is a possibility that these drugs may bind to other receptors due to their multiple target sites. Therefore, other suitable mechanistic investigations backed by computational studies are required that aim to form conclusions on the efficacy of these drug repurposing molecules, which could be needed to fight against this type of cancer.

## Figures and Tables

**Figure 1 medicina-57-01058-f001:**
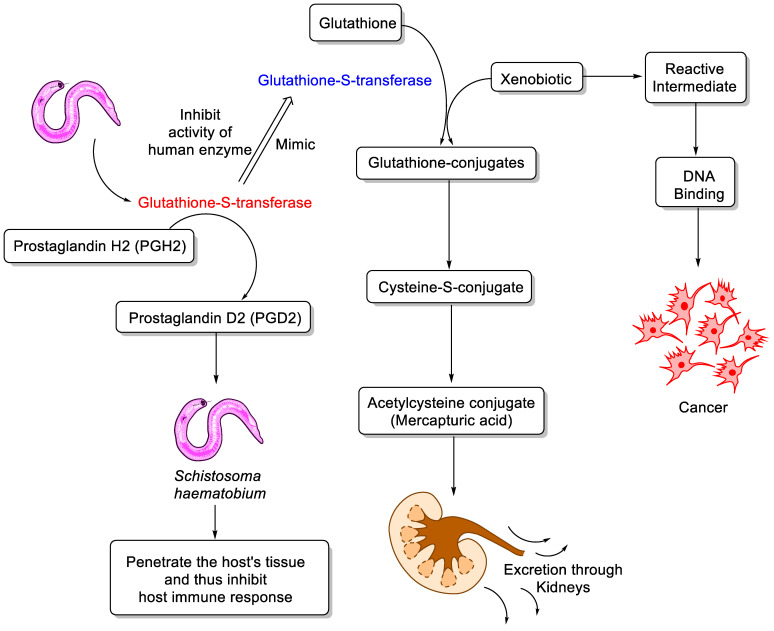
Schematic view and biological role of glutathione S-transferases (GSTs) in *Schistosoma* parasite and cancer.

**Figure 2 medicina-57-01058-f002:**
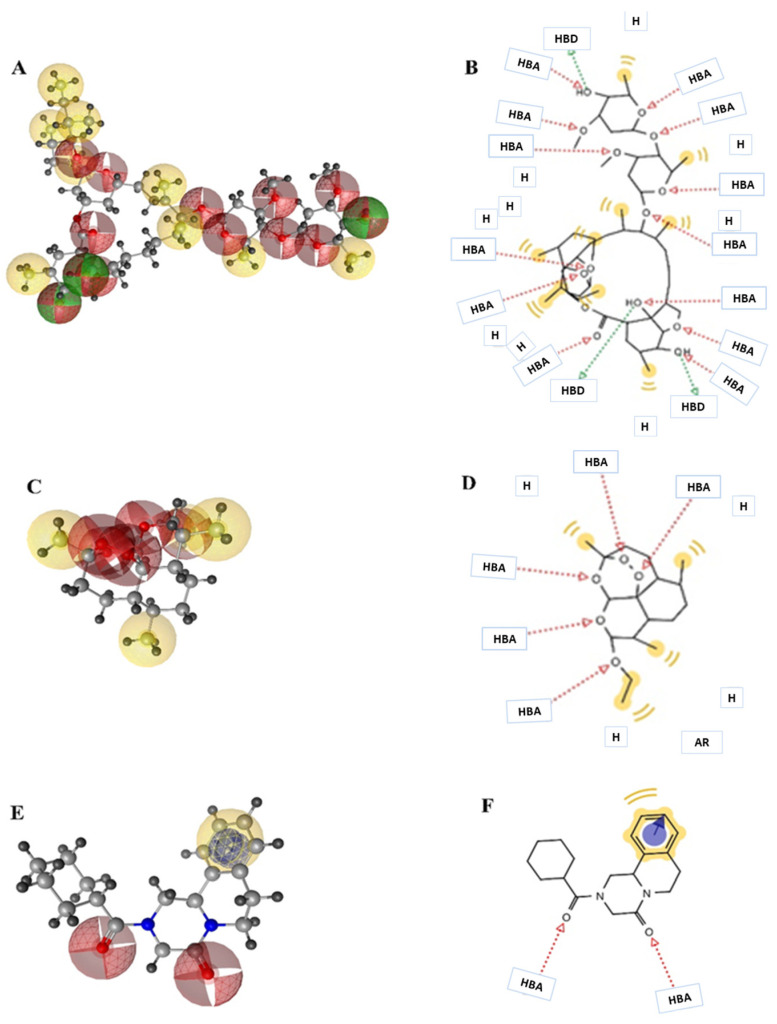
Three-dimensional (**A**,**C**,**E**) and two-dimensional (**B**,**D**,**F**) poses displaying the pharmacophore features of ivermectin, arteether and praziquantel. Green arrows show the H-bond donor (HBD); red arrows show H-bond acceptor (HBA) and yellow colours shows hydrophobic group (H).

**Figure 3 medicina-57-01058-f003:**
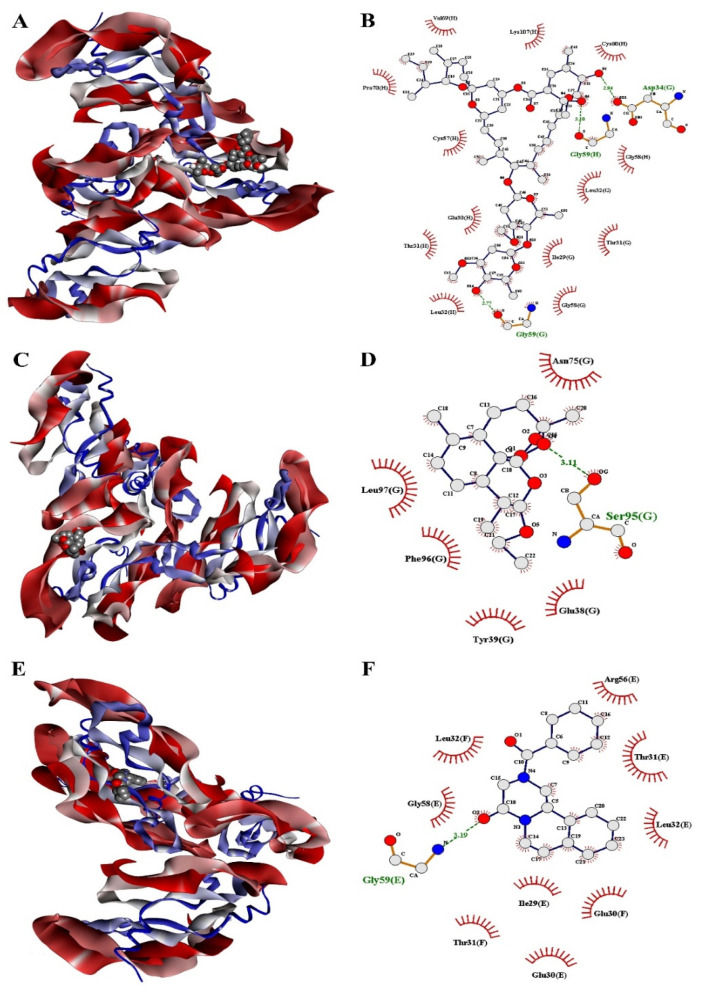
Three-dimensional and two-dimensional docked poses of VEGF–ligand complex for (**A**,**B**) ivermectin; (**C**,**D**) arteether; and (**E**,**F**) praziquantel.

**Figure 4 medicina-57-01058-f004:**
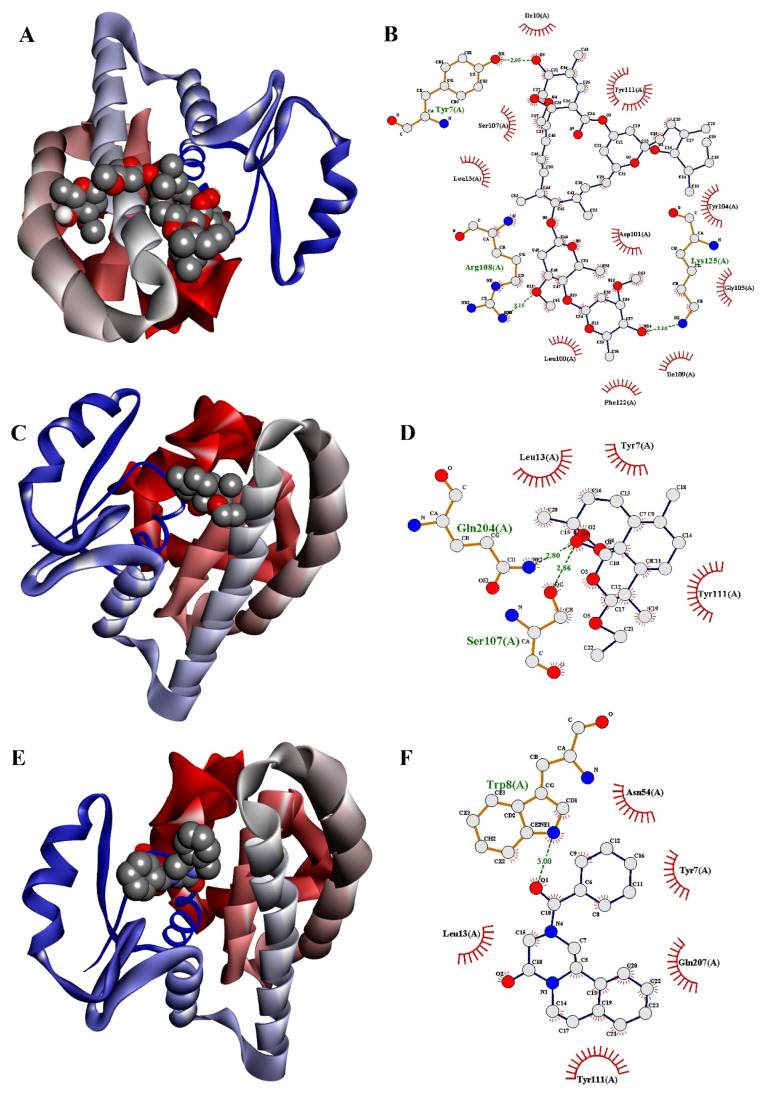
Three-dimensional and two-dimensional docked poses of GST-ligand complex for (**A**,**B**) ivermectin; (**C**,**D**) arteether; and (**E**,**F**) praziquantel.

**Figure 5 medicina-57-01058-f005:**
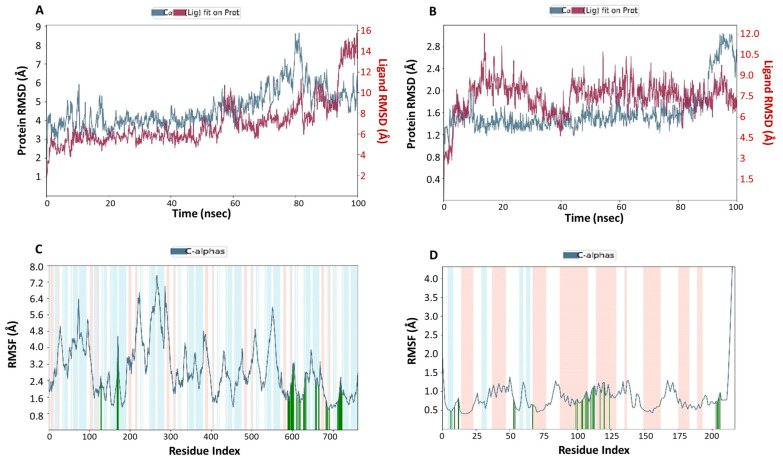
Conformational changes during MD simulation. (**A**,**B**) The plots showing RMSD values of ivermectin–VEGF and ivermectin–GST complexes during 100 ns MD simulation. (**C**,**D**) The plots showing RMSF values of ivermectin–VEGF and ivermectin–GST complexes during 100 ns MD simulation. In RMSF plots, the amino acid residues (interacted with the ligand) are shown by green vertical lines, whereas the salmon and cyan rectangles show the alpha helix and beta-strand domains, respectively.

**Figure 6 medicina-57-01058-f006:**
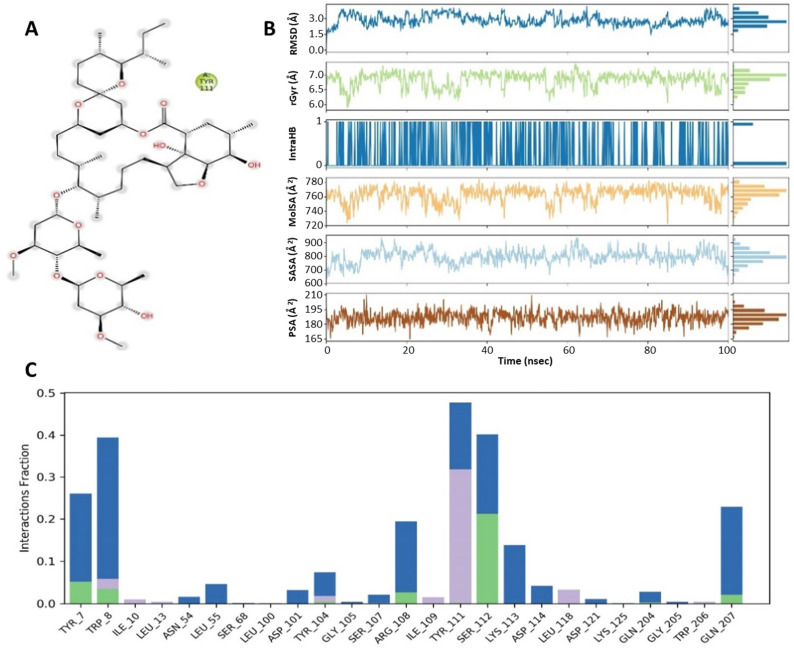
(**A**) Molecular interactions of ivermectin within the pockets of GST receptor at specific sites. (**B**) Demonstration of ivermectin properties as ligand against GST receptor. (**C**) A histogram displaying the different types of interactions within ivermectin–GST complex. The values of RMSD, radius of gyration, molecular surface area, solvent accessible surface area and polar surface area are peaked and represented by the blue line, green line, orange line, cyan blue line and brown line, respectively.

**Figure 7 medicina-57-01058-f007:**
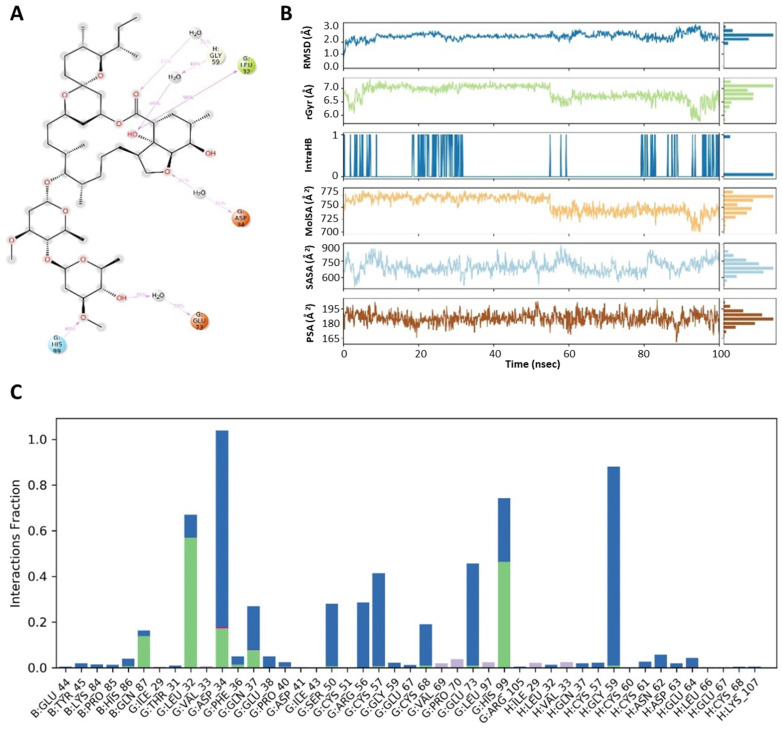
(**A**) Molecular interactions of ivermectin within pockets of VEGF receptor at specific sites. (**B**) Demonstration of ivermectin properties as ligand against VEGF receptor. (**C**) A histogram displaying the different types of interactions within ivermectin–VEGF complex. The values of RMSD, radius of gyration, molecular surface area, solvent accessible surface area and polar surface area are peaked and represented by the blue line, green line, orange line, cyan blue line and brown line, respectively.

**Table 1 medicina-57-01058-t001:** Binding energies of selected drugs with targeted receptors.

Drugs	Proteins (Receptor)	Docking Score	Interacting Amino Acids
Ivermectin 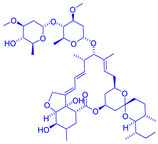	VEGF (1VPF)	−9.2	Ile29, Glu30, thr31, leu32, Cys57, Gly59,Cys 61, Asp63, Glu64
	GST (1GTA)	−9.0	Tyr7, Ile10, Leu13,leu100, Asp101, Tyr104, Gly105, Ser107, Arg108, Ile109, Tyr111, Phe122, Lys125
Arteether 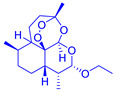	VEGF (1VPF)	−5.5	Asp34, Ser50, Gly59, Cys60, Asp63, Glu64, Glu67, Cys68, Lys107
	GST (1GTA)	−6.0	Tyr7, Ser107, Tyr111, Leu13, Gln204
Praziquantel 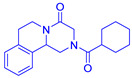	VEGF (1VPF)	−6.1	Ile29, Glu30, Thr31, Leu32, Arg56, Cys57, Gly58, Gly59
	GST (1GTA)	−6.5	Tyr7, Trp8, Leu13, Asn54, Tyr111, Gln207

**Table 2 medicina-57-01058-t002:** MD simulation parameters of ivermectin with VEGF and GST.

Parameters	Complexes	
	Ivermectin–VEGF	Ivermectin–GST
RMSD Cα atoms (Å)	1.957–8.631	1.026–3.041
RMSD ligand fit on protein (Å)	1.99–15.763	2.592–12.065
RMSF Cα atoms (Å)	0.993–7.538	0.401–6.916
rGyr (Å)	5.772–7.321	5.882–7.373
MolSA (Å2)	699.995–785.114	722.788–783.773
SASA (Å2)	514.327–893.285	616.297–937.471
PSA (Å2)	160.604–204.182	164.98–209.46
Hydrogen bond	Gly59, His86, Gln87, Leu32, Asp34, Phe36, Gln37, Ser50, Cys57, Cys68, Glu73, His99	Tyr7, Trp8, Tyr104, Arg108, Tyr111, Ser112, Gln204, Gln207
Hydrophobic bond	Ile29, Val33, Pro40, Cys60, Val69, Pro70, Leu97	Trp8, Ile10, Leu13, Leu100, Tyr104, Ile109, Tyr111, Leu118, Trp206
Ionic bond	Asp34, Glu64	-
Water bridges	Thr31, Leu32, Val33, Asp34, Phe36, Gln37, Glu38, Pro40, Asp41, Glu44, Tyr45, Ser50, Arg56, Cys57, Gly59, Cys61, Asn62, Asp63, Glu64, Leu66, Glu67, Cys68, Glu73, Lys84, Pro85, His86, Gln87, His99, Arg105, Lys107	Tyr7, Trp8, Asn55, Ser68, Asp101, Tyr104, Gly105, Ser107, Arg108, Tyr111, Ser112, Lys113, Asp114, Asp121, Lys125, Gln204, Gly205, Gln207

## Data Availability

All data generated or analyzed during this study are included in this article.

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
