# Peer review of "Molecular Docking and Dynamics Simulation Revealed Ivermectin as Potential Drug against *Schistosoma*-Associated Bladder Cancer Targeting Protein Signaling: Computational Drug Repositioning Approach"

_medicina, 2021, doi:10.3390/medicina57101058_

Round 1

Reviewer 1 Report

In this work, Siddiqui and colleagues describe a drug repurposing approach for Schistosoma associated bladder cancer based on molecular docking and dynamics simulation. The authors suggest ivermectin as a repurposed drug that may affect VEGF and GST signaling in bladder cancer. Although the study provides some interesting findings that worth further evaluation, there are some issues that I would like to bring to the attention of the authors.

  • Although the title of the manuscript implies drug repurposing approach for Schistosoma associated bladder cancer, the proposed drugs can be used also for Schistosoma non-associated bladder cancer. Thus, can be used in general for bladder cancer treatment and this should be clearly stated in the manuscript.
  • It is not clear how GST and VEGF pathways are specifically related to Schistosoma associated bladder cancer. Especially for VEGF it is not clear how is associated with Schistosoma-associated Bladder Cancer and why it was selected as a pathway to be pharmacologically targeted. A better justification for the selection of these drugs and pathways is strongly recommended.
  • According to the introduction praziquantel is a drug already applicable with safety and without toxicity. So why drug repurposing is needed? Due to off-target effects? What are the off-target effects and toxicity of ivermectin and arteether?
  • How does ivermectin work? Which are the targets of this drug and why was selected. The same for artemisinin (arteether). What is the mechanism of action and how are these associated with GST and VEGF?
  • One of the targets of ivermectin is the farnesoid X receptor which is involved in metabolic regulation. Overall, this indicates that there are also multiple targets for ivermectin; thus this should be clearly stated in the limitations. What is the predicted affinity for other receptors? Ivermectin may have higher affinity and make more stable complexes with other receptors as well that may be implicated in BC progression. How can this be excluded?
  • Praziquantel was selected as a reference drug instead of other drugs that specifically target these receptors. Why the authors did not use known inhibitors (non-substrate) for that particular proteins/receptors and instead use praziquantel?
  • What will be the advantage of using these drugs instead of the one already being in the market (praziquantel)?
  • There are several syntax and typographical errors in different sections of the manuscript. A careful proof-reading is required. Also, in some places in the manuscript ivermectin-2VPF is written and in others 1VPF. In other GTA signaling instead of GST. Please have consistency.
  • Please also describe the limitations of the study.
  • In Section 3.2 the authors state that VEGF and GST inhibitors are strongly recommended for cancer treatment. This is a vague statement and please rephrase.
  • Ivermectin and arteether were not confirmed but were predicted as drugs that may target these pathways. Please rephrase lines 469-470. Furthermore, according to the findings arteether has a weak binding to both receptors so how is possible to obstruct the pathways?
  • Several validation studies are needed to prove the efficacy of these drugs prior moving to the clinic. This should be clearly stated in the limitations.

Author Response

Reviewer 1

Comment 1: In this work, Siddiqui and colleagues describe a drug repurposing approach for Schistosoma associated bladder cancer based on molecular docking and dynamics simulation. The authors suggest ivermectin as a repurposed drug that may affect VEGF and GST signaling in bladder cancer. Although the study provides some interesting findings that worth further evaluation, there are some issues that I would like to bring to the attention of the authors.

Response 1:  We thank reviewer for the positive review of our manuscript. Reviewer’s critical and insightful comments definitely led to the improved final version. As per the reviewer’s comment, we have now revised the manuscript accordingly and enhanced articulately the concept and expression of the work.

Comment 2: Although the title of the manuscript implies drug repurposing approach for Schistosoma associated bladder cancer, the proposed drugs can be used also for Schistosoma non-associated bladder cancer. Thus, can be used in general for bladder cancer treatment and this should be clearly stated in the manuscript.

Response 2: Thank you for the comment. Following the reviewer’s comment, we have incorporated the respective statement in the revised manuscript and can be seen as highlighted.

Comment 3: It is not clear how GST and VEGF pathways are specifically related to Schistosoma associated bladder cancer. Especially for VEGF it is not clear how is associated with Schistosoma-associated Bladder Cancer and why it was selected as a pathway to be pharmacologically targeted. A better justification for the selection of these drugs and pathways is strongly recommended.

Response 3: Schistosoma haematobium, a parasitic flatworm that infects more than 100 million people, mostly in the developing world, is the causative agent of urogenital schistosomiasis, and is associated with a high incidence of squamous cell carcinoma of the bladder. During infection, eggs are deposited in the bladder causing an intense inflammatory reaction. A growing amount of evidence points to angiogenesis playing a key role in schistosomiasis-associated bladder cancer as well as in schistosomiasis. This statement may seem a paradox, since schistosomes are intravascular parasites that cause damage by destroying the blood vessels. On the other hand, angiogenesis, or the formation of new endothelial sprouts from preexisting postcapillary venules, is a well-known characteristic of inflammatory diseases, wound repair and provides support for tumor progression. See the references…

https://doi.org/10.1111/apm.12756

DOI: 10.1590/s0074-02762010000400013

VEGF linkage with Schistosoma-associated bladder cancer

A large number of proangiogenic factors including vascular endothelial growth factor (VEGF), basic fibroblast growth factor (bFGF), hepatocyte growth factor (HGF) and angiogenin (ANG) are often overexpressed in tumors Several studies have indicated that angiogenic activators play an important part in the growth and spread of tumors. On immunohistochemical examination, the VEGF family and their receptors were found to be expressed in about half of the human cancers investigated. These factors are known to affect the prognosis of adenocarcinomas that have developed in the uterine cervix, endometrium, ovary and stomach. In addition, a significant correlation between the expression of VEGF and prognosis has been described in colorectal cancer, breast cancer, lung cancer, head and neck squamous cell carcinoma, Kaposi sarcoma and malignant mesothelioma.

https://pubmed.ncbi.nlm.nih.gov/9684805/

DOI: 10.1002/cncr.20449

https://doi.org/10.1111/apm.12756

There are many studies which have demonstrated the direct or indirect role of VEGF as well as GST in Schistosoma-associated bladder cancer. Some studies and their link is mentioned below in detail. Moreover, following the reviewer comment, we have also now justified this part precisely in the manuscript.

Study 1: It has been demonstrated that intact live eggs, excretory/secretory products of eggs and the extracts of homogenized eggs stimulate the proliferation and migration of endothelial cells. Formation of endothelial capillary-like outgrowths, was stimulated by egg extracts. The effects mediated by eggs of schistosomes revealed that the soluble egg antigen induces endothelial cell proliferation and upregulates vascular endothelial growth factor (VEGF).

DOI: 10.1590/s0074-02762010000400013

Study 2: Loeffler et al. investigated the effects of Schistosoma mansoni soluble egg antigen (SEA) on angiogenic processes: proliferation, tube formation and apoptosis of human umbilical vein endothelial cells (HUVECs). In this study, SEA increased HUVEC tube formation and decreased HUVEC apoptosis after serum and growth factor deprivation. These authors showed that messenger RNA for vascular endothelial growth factor (VEGF) increased 2-fold in SEA-treated HUVECs. Their findings suggest that products secreted by schistosome eggs may promote angiogenesis by upregulating endothelial cell VEGF. Other authors analyzed VEGF levels in sera from people diagnosed with schistosomiasis. These patients had significantly high VEGF levels compared with healthy people. Therefore, this angiogenic capacity has been suggested as an early marker of preneoplastic and neoplastic lesions in schistosomiasis associated squamous cell carcinoma.

DOI: 10.1086/340416

https://pubmed.ncbi.nlm.nih.gov/14964670/

https://pubmed.ncbi.nlm.nih.gov/21296079/

Study 3: Another study showed differential expression of genes belonging to cancer-associated and vascular endothelial growth factor (VEGF)-associated pathways, which is consistent with the observation that exposure to S. mansoni soluble egg antigen induces increased production of VEGF in human umbilical vein endothelial cells, as well as the report of elevated serum and urine VEGF protein levels in bladder cancer patients infected with S. haematobium. These pathways provide an important starting point in further investigations to identify parasite and host molecules that give rise to schistosomal bladder cancer

https://www.ncbi.nlm.nih.gov/pmc/articles/PMC6104441/

https://pubmed.ncbi.nlm.nih.gov/12023772/

https://pubmed.ncbi.nlm.nih.gov/23209855/

GST linkage with Schistosoma-associated bladder cancer

The association of bladder cancer with schistosomiasis seems to be related to the endemicity of the parasite and a large amount of evidence now links schistosomiasis of the urinary tract to bladder cancer. Previous studies have shown that reduced glutathione (GSH) level and glutathione S-transferase (GST) activity can reduce the covalent binding of the activated forms of well known carcinogens, e.g. benzo(a)pyrene, NDMA, aflatoxin B1, to DNA and other macromolecules. The higher activity of glutathione S-transferase with some polycyclic aromatic hydrocarbons in many human tissues suggests that GST may play an important role in the detoxification of lung carcinogens. In keeping with these findings, inhibition of GST activity in bladder cancer and in schistosome-infected human bladder cancer tissues may potentiate the binding of alkylating moieties, resulting from bioactivation of N-nitrosamines, with DNA or proteins. Inhibition of GST activity might be due to the presence of certain types of N-nitrosamines in the urine of schistosomal patients since these compounds are well known to decrease such activity.

https://doi.org/10.1016/j.canlet.2003.09.023

https://pubmed.ncbi.nlm.nih.gov/1638515/

https://pubmed.ncbi.nlm.nih.gov/8564926/

Comment 4: According to the introduction praziquantel is a drug already applicable with safety and without toxicity. So why drug repurposing is needed? Due to off-target effects? What are the off-target effects and toxicity of ivermectin and arteether?

Response 4: Praziquantel

Concise version is added in the manuscript.

Because of its efficacy, safety, cost, and indeed the lack of alternatives, PZQ has remained the drug of choice for schistosomiasis treatment and transmission control for >40 years. Yet PZQ has drawbacks, including inactivity against juvenile schistosomes. Moreover, reliance on a single drug for the treatment of a disease with the global public significance of schistosomiasis risks facilitating the development and spread of drug resistance, especially since reduced susceptibility has occurred frequently both in the field and in the laboratory. A pressing need for new interventions has arisen, including novel compounds with modes of action discrete from those of PZQ and methods to detect the appearance and spread of resistance to PZQ. Reduced susceptibility to PZQ has been widely found in foci of endemicity, notably in Africa, including Egypt and Senegal. An extremely low cure rate (18%) was reported in Senegal; however, it was suggested that failure of PZQ therapy occurs because of factors other than drug resistance, including very intense transmission and the presence of PZQ-refractory juvenile worms (immature parasites). In Egypt, eggs obtained from treated and uncured patients gave rise to schistosomes (S. mansoni) that showed 3- to 5-fold lower sensitivity to PZQ. Moreover, PZQ repurposing was not done, instead it was used as comparative control due to its universal multifaceted inhibitory action.

https://www.ncbi.nlm.nih.gov/pmc/articles/PMC5404606/

https://pubmed.ncbi.nlm.nih.gov/3914644/

https://pubmed.ncbi.nlm.nih.gov/11703840/

https://pubmed.ncbi.nlm.nih.gov/8780463/

Ivermectin

Ivermectin proved to be even more of a ‘Wonder drug’ in human health, improving the nutrition, general health and wellbeing of billions of people worldwide ever since it was first used to treat Onchocerciasis in humans in 1988. It proved ideal in many ways, being highly effective and broad-spectrum, safe, well tolerated and could be easily administered (a single, annual oral dose). It is used to treat a variety of internal nematode infections, including Onchocerciasis, Strongyloidiasis, Ascariasis, cutaneous larva migrans, filariases, Gnathostomiasis and Trichuriasis, as well as for oral treatment of ectoparasitic infections, such as Pediculosis (lice infestation) and scabies (mite infestation). 

As mentioned above, the broad-spectrum antiparasitic drug IVM, which is widely used in the field of parasitic control, has many advantages that suggest that it is worth developing as a potential new anticancer drug. IVM selectively inhibits the proliferation of tumors at a dose that is not toxic to normal cells and can reverse the MDR of tumors. Importantly, IVM is an established drug used for the treatment of parasitic diseases such as river blindness and elephantiasis. It has been widely used in humans for many years, and its various pharmacological properties, including long- and short-term toxicological effects and drug metabolism characteristics are very clear. In healthy volunteers, the dose was increased to 2 mg/Kg, and no serious adverse reactions were found, while tests in animals such as mice, rats, and rabbits found that the median lethal dose (LD50) of IVM was 10-50 mg/Kg. In addition, IVM has also been proven to show good permeability in tumor tissues. 

Ivermectin has powerful antitumor effects, including the inhibition of proliferation, metastasis, and angiogenic activity, in a variety of cancer cells. This may be related to the regulation of multiple signaling pathways, programmed cancer cell death, including apoptosis, autophagy and pyroptosis.

https://www.ncbi.nlm.nih.gov/pmc/articles/PMC3043740/

https://pubmed.ncbi.nlm.nih.gov/7814280/

https://www.ncbi.nlm.nih.gov/pmc/articles/PMC7505114/

https://pubmed.ncbi.nlm.nih.gov/31845908/

https://www.frontiersin.org/articles/10.3389/fonc.2021.670804/full

Arteether

Arteether is safe and effective alone or combination with some other antimalarial drugs. No adverse event was found in various studies. One such example is this (https://www.ncbi.nlm.nih.gov/pmc/articles/PMC5571326/). Similarly, Artemotil is the ethyl ether derivative of dihydroartemisinin. It was the choice of the WHO for development and was considered less toxic, because one would expect it to be metabolized to ethanol rather than methanol. It is also more lipophilic than artemether, a possible advantage for accumulation in brain tissues. As first-line antimalarial medicines, ARTs are safe, low-toxic and well tolerable. Furthermore, outcomes from a limited number of clinical trials provide encouraging evidence for their excellent antitumor activities. However, some problems such as poor solubility, toxicity and controversial mechanisms of action hamper their use as effective antitumor agents in the clinic. In order to accelerate the use of ARTs in the clinic, researchers have recently developed novel therapeutic approaches including developing novel derivatives, manufacturing novel nano-formulations, and combining ARTs with other drugs for cancer therapy.

https://doi.org/10.1016/B978-0-444-53717-1.00328-0

https://www.frontiersin.org/articles/10.3389/fphar.2020.529881/full.

https://www.ncbi.nlm.nih.gov/pmc/articles/PMC5571326/

The mechanism of antitumor action of ARTs mainly involved in apoptotic cell death which has been confirmed by most literatures. Recognized endoperoxide bridge pharmacophore could be reduced by heme or free ferrous iron to generate carbon free radical and reactive oxygen species (ROS). Excess production of ROS is known to cause apoptotic cell death. However, new mechanisms of action in antitumor activity of ARTs by affecting non-apoptotic cell death including oncosis, autophagy and ferroptosis were also found. Other multiple hallmark events of cancer development and progression were also affected by ARTs including the suppression of cancer cell proliferation, anti-angiogenesis, anti-cancer metastasis and invasion, induction of cell cycle arrest, disruption of cancer signaling pathway and regulation of tumor microenvironment. In tumor microenvironment, there are four types of cells that inhibit immune function including regulatory T cells (Tregs), myeloid-derived suppressor cells (MDSCs), tumor-associated macrophages and cancer-associated fibroblasts. As a result, tumor-specific T cells are unable to enter tumor tissues or their functions are impaired after entering the tissues. This indicates that inhibiting immunosuppression will be beneficial for cancer therapy. Furthermore, it is well-known that cancer cells are different from normal cells with rapid proliferation and metabolic changes especially glycolytic metabolism. Fortunately, ARTs showed antitumor activity by affecting immunosuppression and cancer metabolism. The pace on the study of mechanisms of action of ARTs still doesn’t stop after receiving these exciting results. Cancer stem cells (CSCs) attract the attention of researchers of ARTs because of their crucial role on tumor occurrence, metastasis and recurrence. Although only a few articles about how ARTs affect CSCs have been reported, these new discoveries might provide a revolutionary approach for cancer therapy.

https://pubmed.ncbi.nlm.nih.gov/19883518/

https://pubmed.ncbi.nlm.nih.gov/29901098/

https://pubmed.ncbi.nlm.nih.gov/28254675/

https://pubmed.ncbi.nlm.nih.gov/27119499/

Comment 5: How does ivermectin work? Which are the targets of this drug and why was selected. The same for artemisinin (arteether). What is the mechanism of action and how are these associated with GST and VEGF?

Response 5: Arteether Mechanism and Possible Link

The mechanism of antitumor action of ARTs mainly involved in apoptotic cell death which has been confirmed by most literatures. However, new mechanisms of action in antitumor activity of ARTs by affecting non-apoptotic cell death including oncosis, autophagy and ferroptosis were also found. Other multiple hallmark events of cancer development and progression were also affected by ARTs including the suppression of cancer cell proliferation, anti-angiogenesis, anti-cancer metastasis and invasion, induction of cell cycle arrest, disruption of cancer signaling pathway and regulation of tumor microenvironment. ARTs further inhibits the glycolysis capacity in various tumor cells. Mi et al. first reported that DHA suppressed glucose uptake and glycolysis in non-small-cell lung carcinoma cells and confirmed the effect associated with inhibiting mTOR activity and reducing glucose transporter 1 (GLUT1) expression. Subsequently, Vatsveen et al. observed that ARS decreased glycolysis capacity and mitochondrial respiration capacity in B-cell lymphoma cells although detailed mechanisms remain to be elucidated. However, inhibition of glycolysis of DHA was illustrated via inhibiting PI3K/AKT pathway, downregulating HIF-1α expression and down-regulating pyruvate kinase M2. The oxidative pentose phosphate pathway, another catabolic pathway of glucose, is important for tumor growth and cancer cell metabolism. Elf et al. (2017) reported that targeting 6-phosphogluconate dehydrogenase could sensitize leukemia cells to DHA in oxidative pentose phosphate pathway. Besides an effect on glycolytic metabolism, ARS inhibited HCT116 colon cancer cell proliferation by suppressing the fatty acid biosynthetic pathway, mainly downregulating three proteins: acyl-CoA synthetase 5, hydroxyacyl-coenzyme A dehydrogenase and fatty acid synthase.

https://pubmed.ncbi.nlm.nih.gov/30557609/

https://pubmed.ncbi.nlm.nih.gov/28526807/

https://pubmed.ncbi.nlm.nih.gov/25799586/

https://pubmed.ncbi.nlm.nih.gov/32546972/

https://pubmed.ncbi.nlm.nih.gov/27270429/

Since their discovery in 1961 in rat liver, GSTs have gained attention among cancer researchers. The expression of GSTs in all cell types and their abundance in aggressive cancer cells suggest that they play a key role in tumor progression and cancer pathogenicity. Recent developments in the field of redox oncology have shed light on novel functions of GST proteins in cancer cells. GSTs have emerged as a promising therapeutic target because specific isozymes are overexpressed in a wide variety of tumors and may play a role in the etiology of other diseases, including neurodegenerative diseases, multiple sclerosis, and asthma. Some of the therapeutic strategies so far employed will be described in this section. An initial approach was the design of GST inhibitors to act as reasonably non-toxic modulatory agents, useful in situations where conventional anticancer agents are detoxified by GSTs. Alternatively, compounds designed to disrupt the protein: protein interactions of GST with stress kinases also provide a possible therapeutic approach with respect to modifying proliferative responses.

https://doi.org/10.3390/antiox10050701

https://www.ncbi.nlm.nih.gov/pmc/articles/PMC6361125/)

https://www.sciencedirect.com/science/article/pii/S131901642100061X

Ivermectin Mechanism and Possible Link

Many in vivo and in vitro data highlight the potential of Ivermectin as an anti-cancer agent against a wide range of cancers. For example, in ovarian cancer, ivermectine causes DNA damage through induction of double-strand DNA breaks and induces intrinsic apoptosis by disrupting the mitochondrial membrane, associated with upregulation of BAX/BCL-2 and cytochrome C release. Similar effects are seen in ivermectin-treated liver cancer. Interestingly, Ivermectin induces cell death in leukemia via a different mechanism of action, such as increasing the intracellular level of chloride ions via plasma-membrane hyperpolarization and increasing ROS levels. Furthermore, Ivermectin inhibits cell proliferation and induces apoptosis in colon cancer cells by blocking the canonical WNT pathway, only on TCF-dependent cell types. In glioblastoma, Ivermectin inhibits angiogenesis and deactivates the Akt/mTOR signaling pathway following mitochondrial stress and enhanced ROS levels. Therefore, our study was designed with a possible assumption supported by various evidences that VEGF and GST pathways are somehow directly or indirectly involved in Schistosoma-associated bladder cancer and can possibly be targeted by these two wonder drugs already approved and under clinical trial for anticancer efficacy.

https://www.ncbi.nlm.nih.gov/pmc/articles/PMC8269327

https://pubmed.ncbi.nlm.nih.gov/27852506/

https://pubmed.ncbi.nlm.nih.gov/27551889/

https://pubmed.ncbi.nlm.nih.gov/27771251/

Comment 6: One of the targets of ivermectin is the farnesoid X receptor which is involved in metabolic regulation. Overall, this indicates that there are also multiple targets for ivermectin; thus this should be clearly stated in the limitations. What is the predicted affinity for other receptors? Ivermectin may have higher affinity and make more stable complexes with other receptors as well that may be implicated in BC progression. How can this be excluded?

Response 6: We are very thankful to the reviewer for giving this constructive comment and we duly agree with this comment. There are so many targets of ivermectin where it shows potent biological actions. In our computational studies to search ivermectin targets, we found that it might act as lead drug molecule to combat various human diseases. Suitable work with mechanism backed by computational data would be included in the other manuscript in future and the limitations have now been incorporated in the revised manuscript.

Comment 7: Praziquantel was selected as a reference drug instead of other drugs that specifically target these receptors. Why the authors did not use known inhibitors (non-substrate) for that particular proteins/receptors and instead use praziquantel?

Response 7: We agree with the reviewer’s comment. To clarify this part, we designed this study completely on the basis of parasite. Cancer associated with the parasite, repurpose drugs (ivermectin and arteether) are also anti-parasitic drugs. Therefore, we included comparative control also anti-parasitic drug instead of monoclonal antibody or drugs like Bevacizumab, Rituximab, trastuzumab, cetuximab which can target VEGF or GST.

Comment 8: What will be the advantage of using these drugs instead of the one already being in the market (praziquantel)?

Response 8: Because of its efficacy, safety, cost, and indeed the lack of alternatives, PZQ has remained the drug of choice for schistosomiasis treatment and transmission control for >40 years. Yet PZQ has drawbacks, including inactivity against juvenile schistosomes. Moreover, reliance on a single drug for the treatment of a disease with the global public significance of schistosomiasis risks facilitating the development and spread of drug resistance, especially since reduced susceptibility has occurred frequently both in the field and in the laboratory. A pressing need for new interventions has arisen, including novel compounds with modes of action discrete from those of PZQ and methods to detect the appearance and spread of resistance to PZQ.

https://www.ncbi.nlm.nih.gov/pmc/articles/PMC5404606/

https://pubmed.ncbi.nlm.nih.gov/3914644/

Comment 9: There are several syntax and typographical errors in different sections of the manuscript. A careful proof-reading is required. Also, in some places in the manuscript ivermectin-2VPF is written and in others 1VPF. In other GTA signaling instead of GST. Please have consistency.

Response 9: Following reviewer’s comment, we have now thoroughly checked the whole manuscript for various syntax and typographical errors.

Comment 10: Please also describe the limitations of the study.

Response 10: Limitations of the study are now described in the conclusion section of the manuscript.

Comment 11: In Section 3.2 the authors state that VEGF and GST inhibitors are strongly recommended for cancer treatment. This is a vague statement and please rephrase.

Response 11: Sentence has been rephrased.

Comment 12: Ivermectin and arteether were not confirmed but were predicted as drugs that may target these pathways. Please rephrase lines 469-470. Furthermore, according to the findings arteether has a weak binding to both receptors so how is possible to obstruct the pathways?

Response 12: We thank the reviewer for this valuable suggestion. In the present, we had taken two anti-parasitic drugs and compared them with praziquantel, an approved drug for schistosomiasis. As a results of this study, we found that ivermectin is more active than arteether against Schistosoma-associated bladder cancer. It may or may not obstruct the possible pathways. Accordingly, in this paper, we have reported ivermectin as potent anti-BC molecule, not arteether as it can be clearly seen that binding is weak.

Comment 13: Several validation studies are needed to prove the efficacy of these drugs prior moving to the clinic. This should be clearly stated in the limitations.

Response 13: Limitations of the study are now described in the conclusion section of the manuscript.

Reviewer 2 Report

The authors have performed a computational study that led to identification of Ivermectin as potential drug against Schistosoma-Associated Bladder Cancer Targeting Protein Signaling. In computational studies, the detailed descriptions of methods are required, so that readers can reproduce the study, when necessary. Also, authors have not mentioned several points, which are critical to this field.

  1. The English of the manuscript should be revised. There are a couple of sentences (line 135-137) which are too long and needs grammatical correction (line 37, line 74 etc).
  2. In the abstract, it is mentioned that “three repurposing drugs ivermectin, 30 arteether and praziquantel were virtually screened”. The sentence needs paraphrasing as the study is not exactly an example of virtual screening.
  3. In the introduction it has been mentioned that in-vitro and in-vivo studies demonstrated that artemisinin has anticancer potential. Are there any evidence available in literature about the anticancer activity of Ivermectin? It would be reasonable to see a description regarding the anti-activity related with Ivermectin and Artmesinine. Even, it is not reported in literature, add few sentences about this.
  4. In material and method section 2.1, please add the details of ligand preparation like how many conformations were generated, the energy minimization approach etc.
  5. Similarly, in section 2.2, please add the details of energy minimization process of protein structure like the force field, H-bond optimization approach etc. Does the structure contain any missing residue, if so, how it was handled?
  6. Please add the citation for Ligand Scout 4.4.5 (Line # 170).
  7. In the result section 3.1, the definition of Pharmacophore Modelling is quite unclear (Line #208). There are many references available in the literature (4.06 - Pharmacophore Modeling: 1 – Methods, Comprehensive Medicinal Chemistry II, Volume 4, 2007, Pages 119-147). The authors are advised to go through theses and add a more appropriate definition of pharmacophore modelling.
  8. In the section 3.2, authors have mentioned that by literature survey, it has been clearly shown that VEGF and GTA inhibitors are strongly recommended for the treatment of human cancers. Please add references for these. Also, please add reference to the next line.

Author Response

We thank reviewer for the positive review of our manuscript. Reviewer’s critical and insightful comments definitely led to the improved final version. As per the reviewer’s comment, we have now revised the manuscript accordingly and enhanced articulately the concept and expression of the work.

Comment 1: The authors have performed a computational study that led to identification of Ivermectin as potential drug against Schistosoma-Associated Bladder Cancer Targeting Protein Signaling. In computational studies, the detailed descriptions of methods are required, so that readers can reproduce the study, when necessary. Also, authors have not mentioned several points, which are critical to this field.

The English of the manuscript should be revised. There are a couple of sentences (line 135-137) which are too long and needs grammatical correction (line 37, line 74 etc).

Response 1: Following the reviewer’s comment, we have now rephrased the respective statements in the revised manuscript and can be seen as highlighted.

Comment 2: In the abstract, it is mentioned that “three repurposing drugs ivermectin, 30 arteether and praziquantel were virtually screened”.  The sentence needs paraphrasing as the study is not exactly an example of virtual screening.

Response 2: Following the reviewer’s comment, we have now rephrased the respective statements in the revised manuscript and can be seen as highlighted.

Comment 3: In the introduction, it has been mentioned that in-vitro and in-vivo studies demonstrated that artemisinin has anticancer potential. Are there any evidence available in literature about the anticancer activity of Ivermectin? It would be reasonable to see a description regarding the anti-activity related with Ivermectin and Artmesinine. Even, it is not reported in literature, add few sentences about this.

Response 3: Many in vivo and in vitro data highlight the potential of Ivermectin as an anti-cancer agent against a wide range of cancers. For example, in ovarian cancer, Ivermectine causes DNA damage through induction of double-strand DNA breaks and induces intrinsic apoptosis by disrupting the mitochondrial membrane, associated with upregulation of BAX/BCL-2 and cytochrome C release. Similar effects are seen in Ivermectin-treated liver cancer. Interestingly, Ivermectin induces cell death in leukemia via a different mechanism of action, such as increasing the intracellular level of chloride ions via plasma-membrane hyperpolarization and increasing ROS levels. Furthermore, Ivermectin inhibits cell proliferation and induces apoptosis in colon cancer cells by blocking the canonical WNT pathway, only on TCF-dependent cell types. In glioblastoma, Ivermectin inhibits angiogenesis and deactivates the Akt/mTOR signaling pathway following mitochondrial stress and enhanced ROS levels. Therefore, our study was designed with a possible assumption supported by various evidences that VEGF and GST pathways are somehow directly or indirectly involved in Schistosoma-associated bladder cancer and can possibly be targeted by these two wonder drugs already approved and under clinical trial for anticancer efficacy.

Involvement of these two drugs are now mentioned in the manuscript and can be seen as highlighted. Furthermore, related references are also now added.

https://www.ncbi.nlm.nih.gov/pmc/articles/PMC8269327

https://pubmed.ncbi.nlm.nih.gov/27852506/

https://pubmed.ncbi.nlm.nih.gov/27551889/

https://pubmed.ncbi.nlm.nih.gov/27771251/

Comment 4: In material and method section 2.1, please add the details of ligand preparation like how many conformations were generated, the energy minimization approach etc.

Response 4: Thanks for your suggestion. We have now added the requested section in methodology following below mentioned references.

Comment 5: Similarly, in section 2.2, please add the details of energy minimization process of protein structure like the force field, H-bond optimization approach etc. Does the structure contain any missing residue, if so, how it was handled?

Response 5: Following reviewer’s comment, we have now added the requested part in the revised manuscript. Furthermore, no structure contained any types of residues.

Comment 6: Please add the citation for Ligand Scout 4.4.5 (Line # 170).

Response 6: Done

Comment 7: In the result section 3.1, the definition of Pharmacophore Modelling is quite unclear (Line #208). There are many references available in the literature (4.06 - Pharmacophore Modeling: 1 – Methods, Comprehensive Medicinal Chemistry II, Volume 4, 2007, Pages 119-147). The authors are advised to go through theses and add a more appropriate definition of pharmacophore modelling.

Response 7: Following reviewer’s comment, we have now added the requested definition in the revised manuscript and can be seen as highlighted.

Comment 8: In the section 3.2, authors have mentioned that by literature survey, it has been clearly shown that VEGF and GTA inhibitors are strongly recommended for the treatment of human cancers. Please add references for these. Also, please add reference to the next line.

Response 8: Done.

Round 2

Reviewer 1 Report

Minor changes might still be needed. Although the authors nicely address the comments in the rebuttal letter, in some cases the information is not satisfactory integrated in the manuscript. Small changes in the introduction and discussion are recommended to better guide and avoid misleading of the reader. In the rebuttal letter, the authors nicely elaborate on the selection of GST and VEGF which is really very helpful and provides good evidence on prioritization of these candidates but in the introduction this is not clearly presented. Especially for VEGF the information presented in the introduction is general knowledge and no association with the disease is described. Also in the discussion you mention about the paradox but there is no clear association. On the other hand, in the letter you describe very nicely that products secreted by schistosoma eggs may promote angiogenesis by upregulating endothelial cell VEGF etc. Could you please summarize the information from the letter on how VEGF and GST can be related to schistosoma associated bladder cancer (response number 3) and incorporate it in the manuscript? Also, another recommendation is to include few sentences in the discussion about Praziquantel, Ivermectin and Arteether in the discussion (responses 4 and 8). Few sentences on the mechanism of action of the selected drugs can be also included (response number 5). This will further support your study and the importance of your findings. Also response number 12 could be mentioned in your manuscript as you find appropriate.

Author Response

We are thankful to the reviewer for his/her thorough and insightful comments, which made this manuscript scientifically sound and very much detailed. We really appreciate the efforts made by the reviewer. Moreover, following the reviewer's round 2 comments, we have now incorporated all suggestive details, mechanisms and other informations regarding drugs. All the incorporated changes can be seen in the revised manuscript as highlighted.

This manuscript is a resubmission of an earlier submission. The following is a list of the peer review reports and author responses from that submission.